# A Modified IHACRES Rainfall-Runoff Model for Predicting the Hydrologic Response of a River Basin Connected with a Deep Groundwater Aquifer

**Iolanda Borzì** [1,*,†], **Brunella Bonaccorso** [1] and **Aldo Fiori** [2]

1   Department of Engineering, University of Messina, Villaggio S. Agata, 98166 Messina, Italy;
    bbonaccorso@unime.it
2   Department of Engineering, Roma Tre University, 00146 Rome, Italy; aldo.fiori@uniroma3.it
*   Correspondence: iborzi@unime.it
†   Currently Visiting Ph.D. Student at Institute of Hydraulic Engineering and Water Resources Management,
    Vienna University of Technology, 1040 Vienna, Austria.

**Abstract:** A flow regime is influenced by the degree of hydrologic connection between surface water and groundwater. As this connection becomes more transient and the basin's runoff response more non-linear, such as for intermittent streams, the need for explicit representation of the groundwater component increases. The present study investigates the connection between Northern Etna groundwater system and the Alcantara river basin in Sicily (Italy). In particular, the upstream part of the basin, whose flow regime is essentially intermittent, is modeled through a modified version of the IHACRES rainfall-runoff model. The structure of the model includes a routing module formulated as a two-store model, with the upper store simulating the quick component of the runoff and recharging the lower store which, in turn, describes the slow component of the runoff and the groundwater extraction and losses. Both stores are conceptualized as simple linear reservoirs, with the lower one that maintains a continuous water balance account of groundwater storage volumes for the upstream basin area with respect to a control cross-section, assumed to be the stream gauging station. The model is calibrated at Moio Alcantara cross-section, where daily streamflow data are available. Model calibration and validation are carried out for the period 1980–1984 and 1986–1988, respectively. A first-order analysis is also performed to assess the sensitivity of model parameters. The adopted configuration is shown to improve model performance with respect to the original IHACRES model, with the proposed formulation able to better capture the interactions between the aquifer and the river.

**Keywords:** hydrologic response; groundwater-fed basin; IHACRES rainfall-runoff model

## 1. Introduction

Groundwater extraction from aquifers that are connected with river systems can alter river hydrologic response by reducing the base flow, or low flow, component of streamflow. This may have adverse consequences for riverine ecosystem health and threaten water resources security [1]. To this end, it is particularly important to understand the connectivity between surface and groundwater resources, so that connected systems could be managed as a single system. Deep knowledge of such interaction is indeed required to effectively implement appropriate water policies, as well as to investigate the role played by climatic variability on the observed impacts on water resources availability.

The degree of coupling between groundwater and surface water systems defines a flow regime and has implications for the model structure needed to predict the hydrologic response of a river basin. In perennial systems, there is a permanent connection between surface water and groundwater,

and good results can be obtained from rainfall-runoff models that do not explicitly represent the groundwater storage. While ephemeral streams are defined as having short-lived flow after rainfall, intermittent streams may maintain flow over some sections even during dry periods, due to the local emergence of the water table over the ground level.

Rainfall-runoff models often fail to simulate the hydrologic connection between surface water and groundwater where it tends to be variable in time and space, as for the intermittent streams. This is, for instance, the case of the Alcantara river basin in Sicily region (Italy), whose upstream is intermittent, while its middle valley is characterized by perennial surface flows, enriched by spring water arising from the big aquifer of the Northern sector of the Etna volcano. Since significant groundwater extraction is mainly located in the upstream, an in-depth knowledge of the aquifer-river interaction in this part of the river basin is fundamental for a proper water resources management.

In a previous study, Aronica and Bonaccorso [2] investigated the impact of future climate change on the hydrologic regime of the Alcantara River basin, by combining stochastic generators of daily rainfall and temperature with the IHACRES rainfall-runoff model under different climatic scenarios, to qualitatively detect modifications in the hydropower potential. In their study, some simplifications to the system configuration have been considered to disregard the contribution of the groundwater component, as the emphasis was on simulating surface runoff only.

With reference to the aquifer-river interaction studies, fully coupled models are usually considered the most appropriate models for water resources management. However, coupled surface-groundwater models present several complexities. One of the main challenges is related to the mathematical representation of the flow and head variability between surface and subsurface systems [3]. The choice of the temporal discretization is critical as well. In fact, surface water models often use small time increments (minutes to days) to capture rapid hydrologic changes, while groundwater models require longer time periods (weeks to months) to simulate slower groundwater movement and solute transport. Other important limitations are the need for a large amount of input data and the time required for model development, calibration, and simulation. Overall, complex models, considering a large number of spatially distributed processes, may be characterized by a high degree of uncertainty associated with model parameters, which may affect the model outputs, thus resulting in lower predictive capability.

On the other hand, an alternative modeling approach for aquifer-river interaction processes can consist of a simple, spatially lumped model, using as few parameters as possible to represent the key identifiable river basin features, such as that achieved by combining a rainfall-runoff model with a simple groundwater model.

In the present study, a modified IHACRES model is proposed to better describe the complex connection between Northern Etna groundwater system and the Alcantara river basin. The adopted modeling approach involves an integrated analysis of the basin response to rainfall through the implementation of a two-store routing module, which allows to simulate the groundwater component of the flow, as well groundwater extraction and losses. The modified version of IHACRES model is calibrated and validated at one of the main cross-sections of the Alcantara river basin, namely Moio Alcantara, where daily streamflow data are available for both model calibration and simulation. The structure of the model also provides the opportunity for dealing with parameters uncertainty when very short and poor-quality data series are available for model calibration and validation [4].

In Section 2, first, the case study used for the model application is illustrated. Then, after a brief description of previous IHACRES-based models addressing groundwater-surface water interactions, the proposed modeling approach is described in details. Section 3 presents the main results of the calibration and validation procedures of the new model, as well as of the sensitivity analysis on model parameters through a first-order method. Finally, conclusions are drawn in Section 4.

## 2. Material and Methods

### 2.1. The Alcantara River Basin

The Alcantara River Basin (see Figure 1) is located in North-Eastern Sicily (Italy), encompassing the north side of Etna Mountain, the tallest active volcano in Europe. The river basin has an extension of about 603 km$^2$. The headwater of the river is at 1400 m a.s.l in the Nebrodi Mountains, while the outlet in the Ionian Sea is reached after 50 km. Table 1 lists the main morphometric and hydrologic characteristics of the entire river basin, as well as of its main sub-basin, at Moio Alcantara.

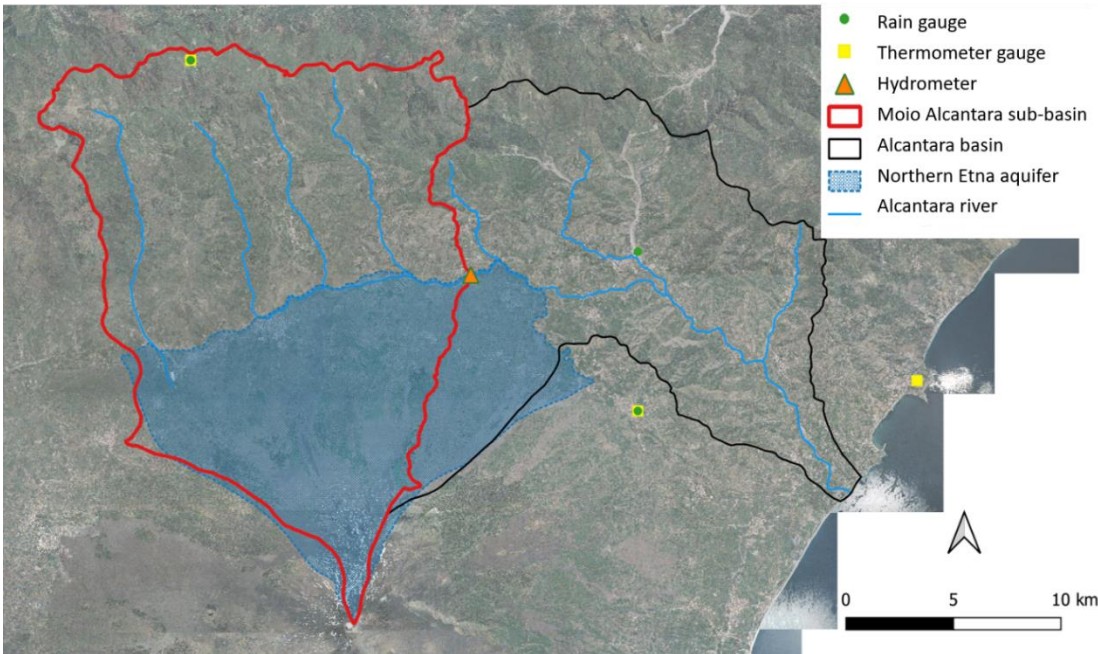

**Figure 1.** Moio Alcantara sub-basin (in red) and the Northern Etna groundwater aquifer (in light blue).

**Table 1.** Main characteristics of the Alcantara Basin and Moio Alcantara sub-basin

| Characteristic | Alcantara Basin | Moio Alcantara Sub-Basin |
|---|---|---|
| Area (km$^2$) | 603 | 342 |
| Mean elevation (m a.s.l.) | 531 | 1142 |
| Max elevation (m a.s.l.) | 3274 | 3274 |
| Min elevation (m a.s.l.) | 0 | 510 |
| Main river length (km) | 54.67 | 34.66 |
| Medium river slope | 0.059 | 0.080 |
| Mean annual rainfall (mm) | 880 | 874 |
| Mean annual runoff (mm) | 342 | 222 |
| Mean annual runoff coefficient | 0.39 | 0.25 |
| Permeable area (%) | 43 | 46 |

On the right-hand side of the river, the mountain area is characterized by volcanic rocks with a very high infiltration capacity and by the lack of a hydrographic network. Here, precipitation and snow melting supply a big aquifer whose groundwater springs at the mid/downstream of the river, mixing with surface water and contributing to feeding the river flow also during the dry season. The left side of the basin is characterized by sedimentary soils (heterogeneous marly clays complex with poor power water-bearing horizons in the rocky levels) where a dense hydrographic network was formed, and gives a seasonal contribution to the river flow, as it follows the rainfall annual variability typical of the Mediterranean climate.

The entire valley hosts many animal species, especially migrating birds. The rich vegetation changes in the different stretches of the river offering a great variety of plans along the way. All this area takes the name of Alcantara Fluvial Park, a wonderful nature reserve which includes a fluvial, a botanical and a geological park.

Groundwater resources, withdrawn in the upper part of the basin (over Moio Alcantara cross-section), are mainly used to supply agricultural areas (see Figure 2) and all the municipalities inside the river basin through local aqueducts, as well as the villages along the Ionian coast, from the Alcantara basin itself up to the city of Messina, by the Alcantara aqueduct [5]. In addition, an existing interconnection between the Alcantara aqueduct and the Messina water distribution system could be used to partially supply the city of Messina, in order to lighten other water sources in the near future. Surface water withdrawals, due to agricultural activities, are mainly concentrated at the downstream, from May to October. Industrial surface water demand is mainly related to paper factories, hydroelectric run-of-river power plants, and mineral extraction. Based on the water balance at the river mouth, about 15% of the annual rainfall recharges the aquifer, primarily during the winter season. The groundwater store has a peak during April-May and then gradually depletes over the summer-fall season until recharge occurs again in winter.

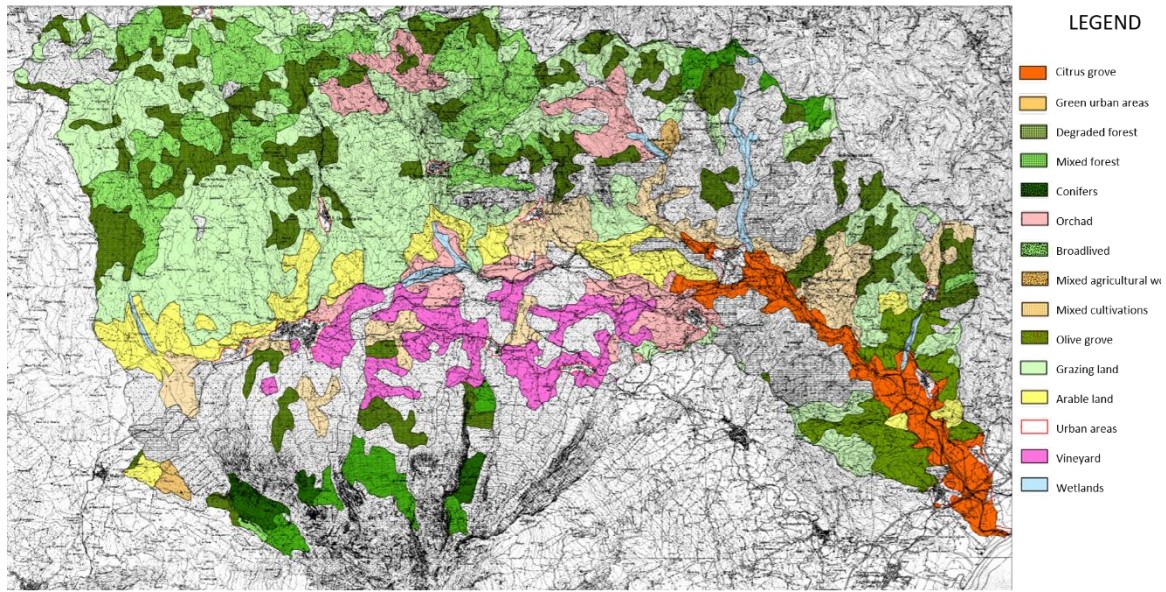

**Figure 2.** Alcantara River Basin land use map.

Beyond the prestigious environmental aspect aforementioned, the Alcantara river basin exhibits several environmental problems due to anthropogenic factors, i.e., urban pressure, industrial settlements (sometimes disused), problems related to flood and landslide defense, and water quality deterioration. Furthermore, following various research studies, climate change effects could exacerbate these criticalities. To this end, 13 municipalities within the Alcantara river basin have agreed upon a River Contract on 22 July 2016, with the aim to define and implement planning tools for the safeguard and the effective management of water resources, the appreciation of fluvial territories and the flood control, and to foster the socio-economic development of the area.

To this end, it is highly important to develop appropriate modeling tools able to simulate the basin's hydrologic response by taking into account the complex aquifer-stream interactions.

### 2.2. The Modified IHACRES Model

The IHACRES rainfall-runoff model proposed by Jakeman and Homberger [6], describes the basin's behavior well in the case that the surface water is the primary component of the flow regime. It is a simple model designed to perform the identification of hydrographs and component flows

purely from rainfall, evaporation, and streamflow data. In this model the rainfall-runoff processes are represented by two modules (see Figure 3): A non-linear loss module that transforms precipitation to effective rainfall, considering the influence of temperature, followed by a linear module based on two parallel transfer functions, represented by exponential equations, which transform the effective rainfall into a quick flow and a slow flow component. The sum of the two components gives the modeled streamflow.

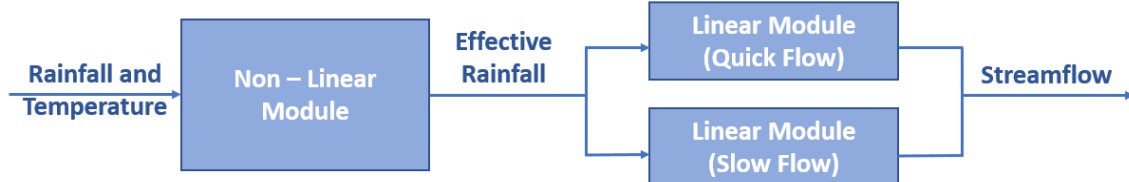

**Figure 3.** IHACRES model structure.

Extensions of the IHACRES model have been proposed by Croke et al. [7], Ivkovic [8], and Herron and Croke [9] to take into account the role of the groundwater component in the hydrologic response of connected groundwater-surface water basins, by appropriately representing the effect of the change in groundwater storage and discharge, also due to water extraction.

Croke et al. [7] integrated a parsimonious, lumped, and physically-based hillslope model, developed by Sloan [10] for homogeneous aquifers, within the IHACRES rainfall-runoff model. The discharge formulation within the groundwater model is expressed as a series of exponential terms and is therefore similar to the commonly used form of the unit hydrograph approach implemented in streamflow models such as IHACRES.

Ivkovic [8] proposed a simple coupled aquifer-river model, entitled IHACRES-GW, where the slow transfer function component of the IHACRES model has been modified by incorporating a groundwater storage module. The latter is conceptualized as a single reservoir, whose areal extent is the basin area upstream of a stream gauging station, considered as the basin outlet. The volume of water released from groundwater storage to the river system is represented by the baseflow component of streamflow. Groundwater extraction and other losses behave as additional outflows from the volume of water held in groundwater storage. The volume of water that recharges the groundwater storage is determined by the proportion of effective rainfall partitioned as slow flow. The remaining fraction of effective rainfall is apportioned to surface runoff. The model was developed for use in unregulated, gauged basins in narrow, semi-confined and narrow, shallow unconfined alluvial valleys with strong aquifer–river connectivity, where groundwater extractions predominantly occur upstream of the gauging station.

Herron and Croke [9] formulated a three-store model (IHACRES-3S), where the slow flow pathway comprises two-layered stores able to capture non-linear hydrologic response better than the linear routing module of the IHACRES-GW model. The upper store receives the volume of effective rainfall partitioned as slow flow, discharges to the stream and recharges the lower store. Conceptually, it can be viewed as a perched water table which develops in response to rain and tends to be relatively short-lived. The lower store corresponds to the groundwater storage in IHACRES-GW.

Keeping in mind the non-linear response of the Moio Alcantara river basin, as well as the specific features of its aquifer system, we modify the original IHACRES model capitalizing on the works by Ivkovic [8] and Herron and Croke [9], in order to properly describe the groundwater discharge and extraction.

The structure of the modified IHACRES model is illustrated in Figure 4. In particular, the routing module is formulated as a two-store model including an upper store to simulate the quick component of the runoff and a lower store that simulates the slow component of the runoff and the groundwater extraction and losses, which is recharged by the first store as well as by the proportion of effective rainfall partitioned as slow flow.

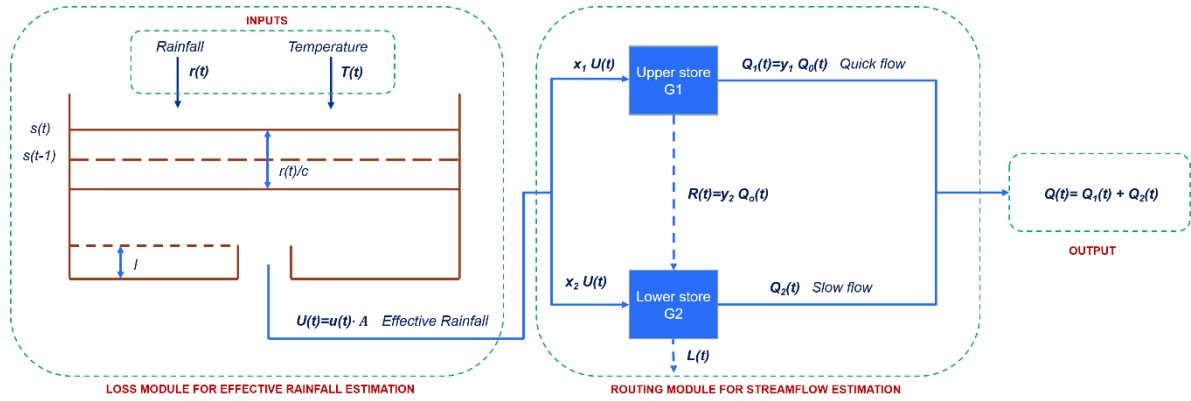

**Figure 4.** Structure of the modified coupled aquifer-river IHACRES model.

Recalling the IHACRES model, the non-linear loss module involves the calculation of an index of basin storage $s(t)$ based upon an exponentially decreasing weighting of total rainfall $r(t)$ and temperature $T(t)$ conditions:

$$s(t) = \frac{r(t)}{c} + \left[1 - \frac{1}{\tau_w(T(t))}\right] \cdot s(t-1) \tag{1}$$

$$\tau_w(T(t)) = \tau_0 \cdot e^{[(20-T(t)) \cdot f]} \tag{2}$$

where $s(t)$ is the basin storage index, or basin wetness/soil moisture index at time $t$, varying between 0 and 1, $\tau_w(T(t))$ is a time constant which is inversely related to the temperature declining rate, $\tau_0$ is the value of $\tau_w(T(t))$ for a reference temperature fixed to a nominal value depending on the climate and usually equal to 20 °C for warmer climates [11], $c$ (mm) is a conceptual total storage volume chosen to constrain the volume of effective rainfall to equal runoff, and $f$ (1/°C) is a temperature modulation factor.

The effective rainfall volume $U(t)$ is finally determined as:

$$U(t) = r(t) \cdot s(t) \cdot A \tag{3}$$

where $A$ is the area of the river basin.

The model assumes that the partitioning of effective rainfall between the two stores is through the constant percentages $x_1$ and $x_2 = 1 - x_1$, respectively. Application of the mass balance equation to the upper store leads to:

$$G_1(t) = G_1(t-1) + x_1 \cdot U(t) - Q_0(t) \tag{4}$$

where $G_1(\cdot)$ is the volume of the upper store and $Q_0(t)$ is the outflow volume at time $t$, which is further portioned into the quick flow component $Q_1(t)$ and the recharge component $R(t)$ through the constant percentages $y_1$ and $y_2 = 1 - y_1$.

Based on the property of linear reservoir, the following relationships between $G_1(t)$ and $Q_0(t)$ hold:

$$Q_0(t) = \begin{cases} a \cdot G_1(t) & if \quad G_1(t) > 0 \\ 0 & otherwise \end{cases} \tag{5}$$

where $a$ is a dimensionless constant equivalent to the storage coefficient for the upper store.

Replacing Equation (5) into Equation (4) results in:

$$G_1(t) = \begin{cases} \frac{1}{1+a} \cdot [G_1(t-1) + x_1 \cdot U(t)] & if \quad G_1(t) > 0 \\ G_1(t-1) + x_1 \cdot U(t) & otherwise \end{cases} \tag{6}$$

Multiplying Equation (6) by *a* allows determining for $Q_0(t)$ a functional form similar to the classical exponential transfer function of the linear routing module in the IHACRES model. Thus, after some algebras, the following relation can be obtained:

$$Q_0(t) = \frac{Q_1(t)}{y_1} = -\alpha_0 \cdot Q_0(t-1) + \beta_0 \cdot U(t) \tag{7}$$

where

$$\alpha_0 = -\frac{1}{1+a} \tag{8a}$$

$$\beta_0 = x_1 \cdot \frac{a}{1+a} \tag{8b}$$

the first parameter related to the rate of the flow recession and the second one related to the height of unit hydrograph peaks of the quick flow. It is worth reminding that $\alpha_0 = -exp\left(-\frac{t}{\tau_0}\right)$, being $\tau_0$ the time constant describing the decay of the outflow from the upper store.

Similarly, the application of the mass balance equation to the lower store leads to:

$$G_2(t) = G_2(t-1) + x_2 \cdot U(t) + R(t) - Q_2(t) - L(t) \tag{9}$$

where $G_2(t)$ is the volume of the lower store, $Q_2(t)$ is the groundwater discharge to the stream and $L(t)$ accounts for both the groundwater extraction and natural losses at time *t*.

In our formulation, the lower store is recharged by the upper store (i.e., $R(t) > 0$) only when $G_1(t) > 0$ and $G_2(t) < 0$, as well as it discharges to the stream (i.e., $Q_2(t) > 0$) only when $G_2(t) > 0$. Following the same line of reasoning as for the upper store, the following relationships between $G_2(t)$ and $Q_2(t)$ can be considered:

$$Q_2(t) = \begin{cases} b \cdot G_2(t) & if \quad G_2(t) > 0 \\ 0 & otherwise \end{cases} \tag{10}$$

where *b* is a dimensionless constant equivalent to the storage coefficient for the lower store.

Replacing Equation (10) into Equation (9) results in:

$$G_2(t) = \begin{cases} \frac{1}{1+b} \cdot [G_2(t-1) + x_2 \cdot U(t) + R(t) - L(t)] & if \quad G_2(t) > 0 \\ G_2(t-1) + x_2 \cdot U(t) + R(t) - L(t) & otherwise \end{cases} \tag{11}$$

Once again, multiplying Equation (11) by *b* allows determining for $Q_2(t)$ a functional form similar to the classical exponential transfer function for the slow component, although without the recharge and groundwater extraction and loss terms, with parameters:

$$\alpha_2 = -\frac{1}{1+b} \tag{12a}$$

$$\beta_2 = x_2 \cdot \frac{b}{1+b} \tag{12b}$$

the first parameter related to the rate of the flow recession and the second one related to the height of unit hydrograph peaks of the slow flow, with $\alpha_2 = -exp\left(-\frac{t}{\tau_2}\right)$, being $\tau_2$ the time constant describing the slow flow decay from the lower store.

Finally, the streamflow discharge is given by:

$$Q(t) = Q_1(t) + Q_2(t) = y_1 \cdot Q_0(t) + Q_2(t) \tag{13}$$

## 3. Results and Discussion

### 3.1. Calibration and Validation of the Modified IHACRES Model

The modified IHACRES model has been applied to the case study described in Section 2.1. The model has in principle a total number of eight independent parameters: Three parameters in the loss module ($\tau_0$, f, c) and five in the routing module ($x_1$, $\tau_1$, $y_1$, $\tau_2$, $L(t)$ ). However, since the overall groundwater extraction in the river basin roughly amounts to 47.5 Mm$^3$ per year, most of which is concentrated above Moio, and that about 32 Mm$^3$ per year are for municipal water use only, while the remaining is mainly for irrigation purpose [5], we have derived a time series for $L(t)$. In particular, 2/3 of the total extracted volume was equally distributed during the year, and the remaining added to the irrigation season, lasting from May to October. These values are assumed to also include the natural losses from the aquifer.

Initially, the input data used for running the model were daily point rainfall and temperature data spatially averaged over the considered area. In particular, the influence of elevation has been taken into account to assess the average daily temperature by means of monthly linear regressions. Clearly, this influence also reflects on the amount of precipitation. On the other hand, no direct relationship between rainfall and elevation has been considered since, at several hundred meters above of the sea level, as the temperature gets cooler with altitude, the maximum precipitable moisture decreases drastically, so that the rainfall-altitude curve reflexes back upon itself. Therefore, we have preferred to use standard weighted Thiessen polygons approach to calculate areal daily rainfall, despite the poor spatial coverage of meteorological stations (see Figure 1).

The model has been calibrated on a four-year daily streamflow discharge time series (1980/81–1983/84) at Moio Alcantara hydrometric station. The calibration period starts in October, at the beginning of the hydrological year, to set the initial condition of the soil storage index to zero. Model calibration has been manually carried out based on visual inspection of modeled versus observed streamflow data, $Q_{obs}$, as well as by minimizing the relative bias (*RB*), namely:

$$RB = \frac{\sum_{t=1}^{n}[Q_{obs}(t) - Q(t)]}{\sum_{t=1}^{n} Q_{obs}(t)} \tag{14}$$

and by maximizing the Nash-Sutcliffe Efficiency (*NSE*), that is:

$$NSE = 1 - \frac{\sum_{t=1}^{n}[Q(t) - Q_{obs}(t)]^2}{\sum_{t=1}^{n}\left[Q_{obs}(t) - \overline{Q_{obs}}\right]^2} \tag{15}$$

where $\overline{Q_{obs}}$ is the mean of the observed streamflow.

In Table 2 the parameter values of the modified IHACRES and the corresponding performance indicators *RB* and *NSE* are reported. Figure 5a shows the comparison between the observed and modeled streamflow for the calibration period, by having considered climate data for effective rainfall generation through the loss module.

**Table 2.** Parameter values and performance indicators of the modified IHACRES model for the calibration period (effective rainfall time-series generated from climate by means of the loss module).

| Parameters | Value | Performance Indicator | Value |
|---|---|---|---|
| $c$ [mm] | 576.70 | | |
| $\tau_0$ | 2 | | |
| $f$ [1/°C] | 3.5 | | |
| $x_1$ | 0.15 | $RB$ | 0.25 |
| $\tau_1$ [days] | 0.46 | $NSE$ | 0.60 |
| $y_1$ | 0.75 (if $G_2(t) < 0$) 1 (if $G_2(t) > 0$) | | |
| $\tau_2$ [days] | 17.63 | | |

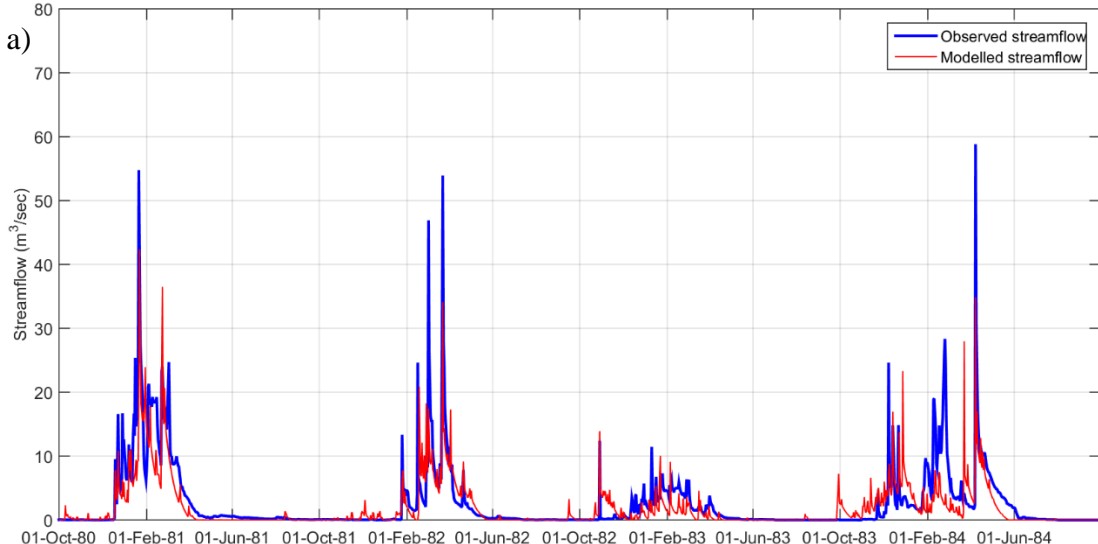

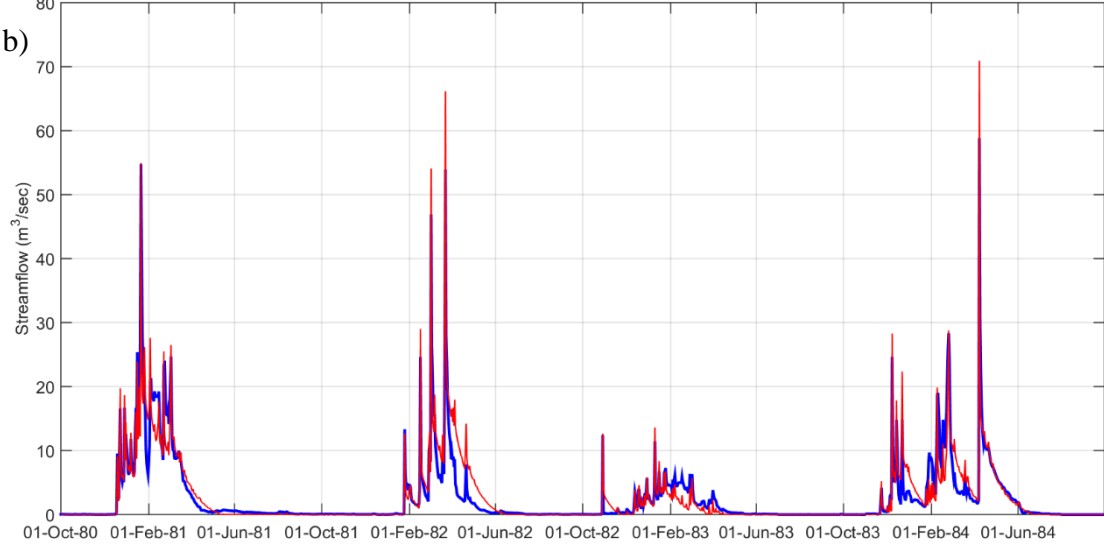

**Figure 5.** Observed versus modeled streamflow for the calibration period, by using as model input (**a**) effective rainfall generated from climate data, or (**b**) effective rainfall generated from observed streamflow.

Despite the relatively good values of the performance indicators, it can be observed that flow peaks are not properly represented. In particular, the resultant modeled streamflow time-series appears to generally under-predict the frequency of quick flow events; besides in some cases modeled peaks occur when there is no observed peak flows. This might be due to the poor spatial coverage of

the meteorological stations, which is not able to represent the non-uniform rainfall pattern over the basin. To work around this issue, following Ivkovic [8], the effective rainfall time-series has been generated from the observed streamflow record, by using a baseflow filter to separate its quick and slow components. First, a running filter of width equal to five-time steps was applied whereby, at each time step t, the minimum of the observed flows was determined. The resulting series is then smoothed using a running average filter of the same width. The filtered series, representing the baseflow contribution to streamflow, was then subtracted from the total streamflow data, yielding the quick flow contribution to streamflow. The effective rainfall was then calculated as:

$$U(t) = \begin{cases} \frac{Q_0'(t) + \alpha_0 \cdot Q_0'(t-1)}{\beta_0} & if \quad Q_0'(t) > Q_0'(t-1) \\ 0 & otherwise \end{cases} \tag{16}$$

where $Q_0'(t)$ is the filtered quick flow at time step $t$.

Therefore, a new calibration has been carried out yielding a new set of model parameters for the routing module (see Table 3). Figure 5b illustrates the comparison between the observed and modeled streamflow by having considered effective rainfall derived through Equation (16).

**Table 3.** Parameter values and performance indicators of the modified IHACRES model for the calibration period (effective rainfall time-series generated from streamflow record).

| Parameters | Value | Performance Indicator | Value |
|---|---|---|---|
| $x_1$ | 0.15 | RB | −0.036 |
| $\tau_1$ [days] | 0.45 | NSE | 0.86 |
| $y_1$ | 0.75 (if $G_2(t) < 0$) 1 (if $G_2(t) > 0$) | RB_s | −0.21 |
| $\tau_2$ [days] | 28.54 | NSE_s | 0.64 |

As expected, Figure 5b shows a better agreement between the observed and modeled streamflow with respect to Figure 5a, with special reference to the timing and, to a large extent, the values of the peaks. This good match is confirmed by the performance indicators for the total streamflow reported in Table 3. In addition, in order to highlight the performance of the model in simulating the aquifer-river interactions, *RB* and *NSE* have also been calculated for the slow flow component, by replacing the observed and modeled total streamflow with the observed (filtered) and modeled baseflow in Equations (14) and (15). The values of the relative bias and the Nash-Sutcliffe efficiency for the slow flow, called respectively *RB_s* and *NSE_s* in Table 3, suggest that the model is also able to capture the recession volumes of baseflow satisfactorily on a daily time step, although the volume of baseflow predicted over the calibration period is about 20% greater than the volume corresponding to the filtered observed flow.

For the validation of the model, the daily streamflow discharge time series observed at Moio Alcantara hydrometric station during the period between October 1986 and September 1988 are used. Once again, effective rainfall generated by observed streamflow has been considered for the validation, to avoid that uncertainty associated with model inputs can be transferred through the model outputs, resulting in lower predictive capability. Results are shown in Figure 6a.

Again, the modeled time series seems to capture the timing of observed peak discharges satisfactorily. However, a closer inspection reveals that the model does not reproduce all parts of the flow regime equally well. In fact, the overall predictive capability of the model deteriorates in simulating the recession curves, as confirmed by the values of the performance indicators in Table 4. Nonetheless, the efficiency of the model can still be considered satisfying also with respect to the simulation of slow flow component, since *NSE_s* is greater than 0.5 [12,13]. In addition, the value of *RB_s* reveals that the error in the assessment of the baseflow volume decreases in absolute terms in comparison to the calibration.

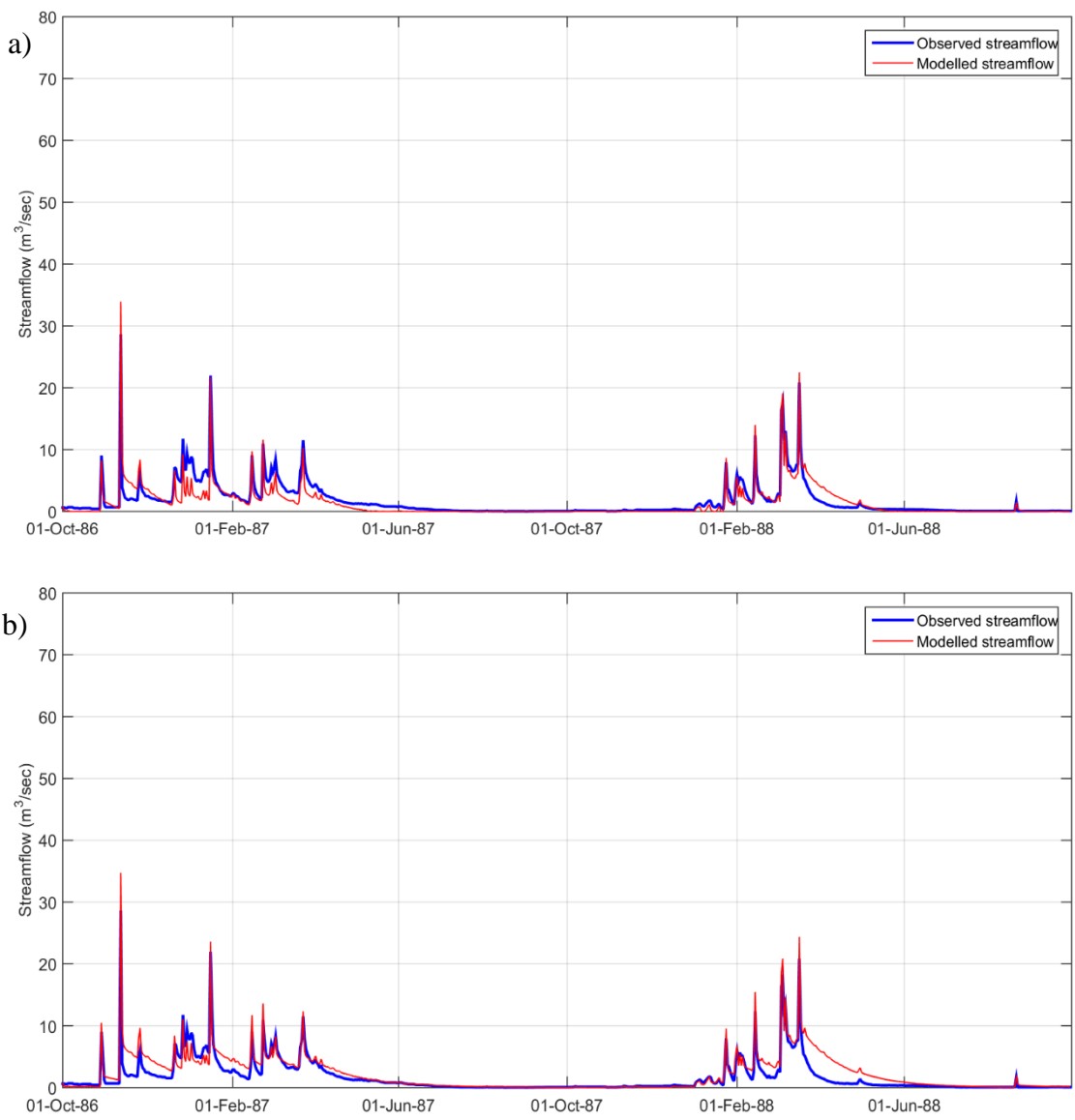

**Figure 6.** Observed versus modeled streamflow for the validation period by means of (**a**) the modified IHACRES model and (**b**) the original IHACRES model.

**Table 4.** Performance indicators of the modified IHACRES model for the validation period.

| Performance Indicator | Value |
|:---:|:---:|
| $RB$ | 0.19 |
| $NSE$ | 0.81 |
| $RB\_s$ | 0.14 |
| $NSE\_s$ | 0.54 |

Finally, in order to objectively judge the added value of the proposed model, the original IHACRES model with two simple linear reservoirs working in parallel with no recharge and no groundwater extension has been run should. The results for the validation period are shown in Figure 6b, while the performance indicators are listed in Table 5. It is evident that the proposed modified IHACRES outperforms the traditional IHACRES in reproducing the interaction between surface water and groundwater.

**Table 5.** Performance indicators of the original IHACRES model.

| Performance Indicator | Calibration Value | Validation Value |
|:---:|:---:|:---:|
| *RB* | −0.34 | −0.31 |
| *NSE* | 0.81 | 0.77 |
| *RB_s* | −0.65 | −0.56 |
| *NSE_s* | 0.47 | 0.20 |

*3.2. Sensitivity Analysis of Modified IHACRES Model Parameters*

In order to understand and assess the sensitivity of the model on its parameters, a first-order analysis was carried out. First-order methods estimate uncertainty in model output assuming that the effects of the individually varying parameters contribute positively to overall model uncertainty. Using an extension of sensitivity analysis, first-order methods predict model variability as a sum of the parameter's variances [14].

In this study, the first-order equation for quantifying system variance is used in its traditional form and it is applied to the calibrated parameters of the routing module ($\tau_1$, $\tau_2$, $x_1$, $y_1$), by:

$$\left[\sigma^2\right]_Q = \sum_{i=1}^{n}\left\{\left[\sigma^2\right]_i \cdot \left[\frac{\delta Q}{\delta i}\right]^2\right\} \tag{17}$$

where $Q$ represents the dependent variable, in our case the total streamflow discharge at Moio cross-section, $\sigma^2$ is the variance of Q, n represents any independent model parameter, that is $\tau_1$, $\tau_2$, $x_1$, $y_1$ and $\frac{\delta Q}{\delta i}$ is the partial derivate of the dependent variable $Q$ with respect to each parameter $i$. This method gives an overall assessment of model sensitivity on its parameters.

In Figure 7 a short subsample of $Q(t)$ (200 days), together with its range of variation, is shown.

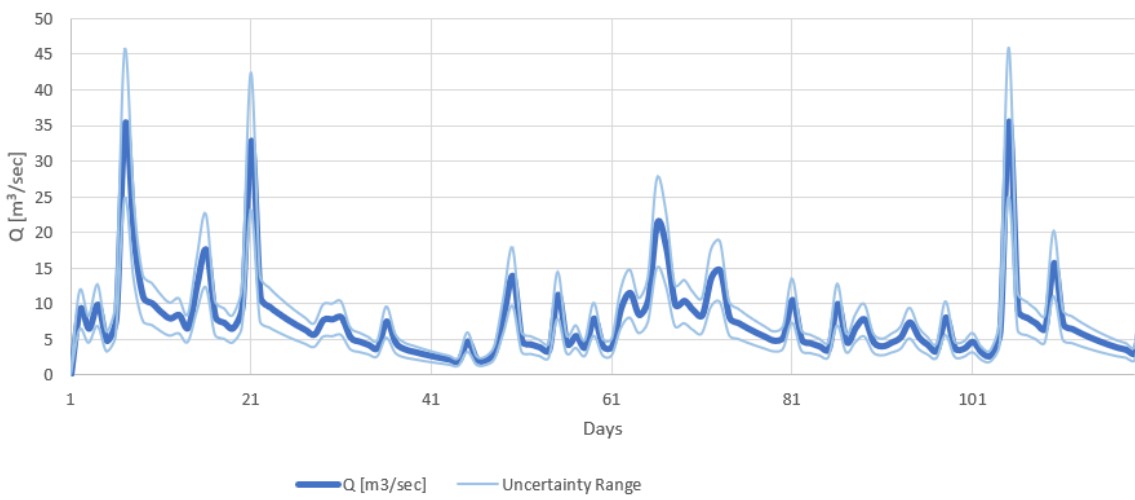

**Figure 7.** Uncertainty analysis in the assessment of the total discharge Q(*t*) based on first-order sensitivity analysis on model parameters.

To better understand how each parameter influences the model's output, another analysis is carried out. As proposed by Sobol [15], for each parameter *i* a first-order sensitivity index or Sobol Index $[S]_i$ is calculated:

$$[S]_i = \frac{\left[\sigma^2\right]_i}{\left[\sigma^2\right]_Q} \tag{18}$$

The value of this first-order sensitivity index for each parameter is reported in Table 6 and a graphic representation of Sobol index values is shown in Figure 8. Sobol indices provide information

about the sensitivity of the model on each parameter. From Table 6 and Figure 8 it can be observed that the model shows a very low sensitivity on parameters $\tau_2$, and $y_1$. Conversely, the model appears more sensitive to the variation of parameters $x_1$, and $\tau_1$, representing the share-out parameter of the effective rainfall $u(t)$, and the storage constant of the quick flow conceptual reservoir. This result highlights how it is important to deeply understand the main physical characteristics of the basin under investigation to keep under control the model simulation dynamics and outputs and to reduce the uncertainties in simulations due to model parameters estimation. To this end, parameter estimation could benefit from the availability of other measured data, such as spring discharges, soil moisture, as well as of a separate calibration to reduce model uncertainties.

**Table 6.** First-order sensitivity index (or Sobol Index) value for each parameter.

| Sobol Index | Value |
|---|---|
| $[S]_{\tau_1}$ | 0.0422314 |
| $[S]_{\tau_2}$ | 0.016778 |
| $[S]_{x1}$ | 0.063349 |
| $[S]_{y1}$ | 0.03130456 |

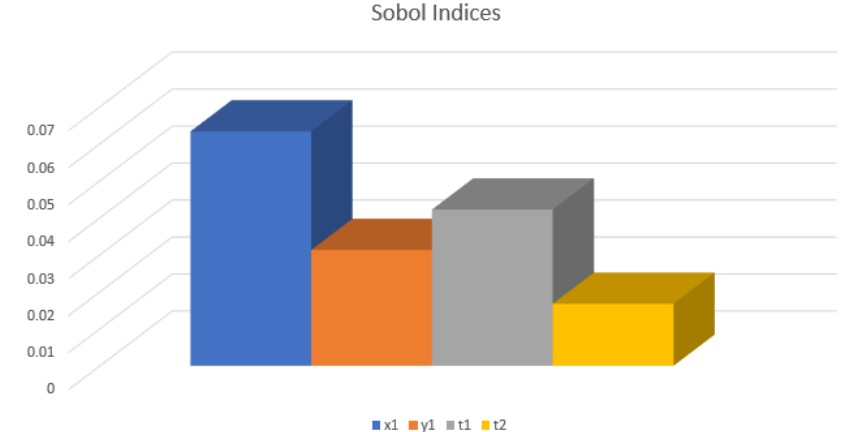

**Figure 8.** Graphical representation of Sobol Index value for each model parameter.

## 4. Conclusions

A modified version of the IHACRES rainfall-runoff model has been proposed to simulate the hydrologic connection between surface water and groundwater in intermittent streams. The proposed model has been developed in the Moio Alcantara river basin, whose groundwater component is associated to a large deep aquifer which may have a critical influence on soil moisture, especially during long dry periods.

The use of a spatially lumped conceptual model, which includes an explicit representation of the interaction of the deep groundwater with the river and the soil water storage, allows to limit the number of parameters necessary to represent the key identifiable river basin features.

The results presented in this paper show improvements in model performance with respect to the original version of the IHACRES model. In particular, the model appears capable to better simulate not only the flood peaks but also the recession curves describing the groundwater aquifer contribution at the end of the wet seasons.

Although a closer inspection reveals some over- or under-estimates of flow, overall the results are encouraging. To this end, it's worth underlining that the simulation is carried out at the daily time scale. However, in many applications, the main concern is in modeling monthly streamflow rather than daily discharges, which usually leads to higher values of the performance indicators.

A first-order sensitivity analysis, carried out to assess the sensitivity of the model on its parameter, reveals a stronger influence of the parameters $x_1$, and $\tau_1$, representing respectively the share-out parameter of the effective rainfall $u(t)$, and the storage constant of the quick flow conceptual reservoir.

The conversion from a two exponential stores in parallel in the original IHACRES configuration, to a two interconnected stores model, with the lower groundwater store recharged by a constant proportion of the effective rainfall and partly by the upper store when the water table is below the elevation of the stream bed (i.e., $G_2(t) < 0$), and depleted by discharges to the stream, extractions and other natural losses involve some assumptions about the system. More specifically, we have assumed that the recharge between the upper and lower stores, $R(t)$ increases linearly with increasing $G_1(t)$, through the parameter $y_2 = 1 - y_1$. An alternative approach consists in deriving for $R(t)$ a soil moisture threshold, g1, below which recharge to the deeper aquifer decreases [9]. However, this solution, as well as other potential refinements of the model parameters to improve the model performance, clearly requires a greater knowledge of the river basin properties than the one that we currently have.

Overall, the proposed model can be considered as a first relevant step towards the implementation of relatively simple conceptual models, easier to use than complex, parameter intensive models, for an effective water resources management in deep groundwater-fed basins.

It should be stressed that sufficient calibration data are required for a valid representation of the connection among deep groundwater storage, soil water storage, and surface runoff. In particular, it's deemed necessary to have measures of spring discharges and soil moisture for proper calibration and implementation of these type of models to predict the basin hydrologic response into the future. Once that these measures will be available, it will be possible to extend the application of the proposed model to the whole Alcantara basin, in order to properly simulate the effect of the interaction between surface water and groundwater at the downstream of the basin.

Future studies will also investigate the possibility to apply this model to other case studies with similar characteristics and with more available data, in order to test the model performance thoroughly.

**Author Contributions:** Model conceptualization, data analysis and visualization, B.B. and I.B.; Writing original draft preparation, review and editing, I.B. and B.B.; Supervision, A.F. and B.B.

**Acknowledgments:** This study was undertaken by the first author as part of the joint Ph.D. Program in Civil, Environmental and Safety Engineering (Cycle XXXII) of the Mediterranean University of Reggio Calabria and the University of Messina. We are grateful to Giuseppe T. Aronica for sharing his knowledge of the IHACRES rainfall-runoff model with us and for his valuable comments on the model development.

**Conflicts of Interest:** The authors declare no conflict of interest.

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
