# Peer review of "A Modified IHACRES Rainfall-Runoff Model for Predicting the Hydrologic Response of a River Basin Connected with a Deep Groundwater Aquifer"

_water, doi:10.3390/w11102031_

Round 1

Reviewer 1 Report

Dear Authors,

You article on the development and application of a modified IHACRES RRM to capture the intermittend flow characteristics of the Alcantara river is interestesting and relevant to the IHACRES and partly the wider modelling community. Besides the description of the model extension the paper shows a calibration and validation to the Alcantara River, and a model parameter uncertainty study. In general, the paper was nicely and consensly written.

However, I have two major concern with your study:

The model performance, especially for the peak discherage that are mainly a result of the fast flow paths (and hence not your extension), is not very good. Discharge peaks in the validation period are in both years not correctly represented. This might be an issue of your precipitation input and needs to be resolved before one can interpret and understand the parameter uncertainty analysis (hence I just briefly went through the sensitivity section).

To be able to objectively judge the added value of the groundwater extension (the aim of the model extension), a (standard) IHACRES model without the new groundwater extension should be presented alongside the newly developed model and the observations. This needs to be added.

Further smaller concerns are:

Figure 1: I have issues in reading and understanding this figure. There are several black lines, but I doubt that these are all catchments. What does the yellow highlighted area indicate? What indicates the black dot in the centre? This might be also a good way to represent the met stations.

Figure3: The parameters in the figure should be explained

Page 8, line 261ff: Depending on the spatial distribution of the met stations, this is a coarse approach, especially given the altitudinal stretch of the catchment. Did you use elevation bands? Please provide more information about the spatial coverage and discuss possible effects on the results. For instance, one could argue that the need for a more sophisticated groundwater model is due to the shortcomings of the input data. This argumentation needs to be disproved.

Page 8, line 262: Indicate, if you use any model-warmup period?

Page 9, line 278: I disagree that the precipitation pattern is nicely represented. Neiter in 1987 nor in 1988 the discharge peaks are captured, indicating that the precipitation representation is not correct. Hence, a more sophisticated precipitation representation is required.

Pag 9, line 292ff: The relatively high NSE values are propably a result of the long period hardy any/without any discharge. performance values should be shown for the "wet" and "dry" periods.

Reviewer 2 Report

The manuscript titled "A Modified IHACRES Rainfall-Runoff Model for Predicting the Hydrologic Response of a River Basin Connected with a Deep Groundwater Aquifer" provided modified rainfall-runoff model including a new groundwater module. The result showed that the necessity of the new module for their study area and acceptable level of model performance. This paper highlights the novality with its design. However, the importance of the study needs to be enriched, and the method issues should be addressed. Some general and specific comments are listed as follows:

General comments:

I think section 2.2 should be in introduction section since it is talking about background, not materials and methods.

You stated that your work work is necessary based on the reference in line 138 which are very old, I wouldn't say they have emerged as a promising strategy. If you can find some new reference support you work that would good. Otherwise, you might just focus on that several studies using this model already have been applied in your study area.

Compared to Figure 3 you showed, I don't see the novality of your work since; first, why there is no groundwater extraction and other losses? why do you except for groundwater losses that represent the aquifer recharge as you said in line 219? I couldn't see the improvement of your work from Ivkovic although you stated his one has limitations. Base on your results, I think you should compare your results with with the previous versions of the model to prove the necessity of your work.

There are some varibales that were explained in your equations and you should point out what has been modified from previous ones.

The results alone can hardly demonstrate the novality or necessity of your work and also the future worked you suggested since the work from one gauged station to others, even the neighborh watershed, need to carefully applied.

Specific comments:

You started to use basins and then changed to catchments, keep consistent

Line 56: qualitatively

Line 63: streamflow as input?

Line 94: maybe show different land use such as urban, industrial, and agricultural

Line 100: reference

Line 104: this figure is very confusing. first, use different colors for Alcantara Basin boundaries; second, there are two separate river with the same symbol, change them; third, which basin this outlet is corresponded with; if you not use Alcantara basin, why do you show it?

Table 1: Main river length for Alcantara Basin: 54.67?

Line 141: simpler than what?

Line 240: I assumed "A" is area and in what unit?

Line 245: explain tr?

Line 249; explain y1?

Line 252: explain tb?

Line 264: show calibrated results in figure

Figure 5: separately name 5a and 5b, you did not mention the upper part of figure 5 so I don't know why you showed it here.

Figure 6: m3/s; give legends to each colored lines

Round 2

Reviewer 2 Report

I accept this version with the edits.